# Presynaptic Vesicle Protein SEPTIN5 Regulates the Degradation of APP C-Terminal Fragments and the Levels of Aβ

**DOI:** 10.3390/cells9112482

**Published:** 2020-11-15

**Authors:** Mikael Marttinen, Catarina B. Ferreira, Kaisa M. A. Paldanius, Mari Takalo, Teemu Natunen, Petra Mäkinen, Luukas Leppänen, Ville Leinonen, Kenji Tanigaki, Gina Kang, Noboru Hiroi, Hilkka Soininen, Kirsi Rilla, Annakaisa Haapasalo, Mikko Hiltunen

**Affiliations:** 1Institute of Biomedicine, University of Eastern Finland, 70210 Kuopio, Finland; mikael.marttinen@uef.fi (M.M.); kaisa.paldanius@crl.com (K.M.A.P.); mari.takalo@uef.fi (M.T.); teemu.natunen@uef.fi (T.N.); petra.makinen@uef.fi (P.M.); luukasl@student.uef.fi (L.L.); Kirsi.rilla@uef.fi (K.R.); 2Instituto de Medicina Molecular—João Lobo Antunes, Faculdade de Medicina, Universidade de Lisboa, 1649-028 Lisboa, Portugal; catarina.ferreira@medicina.ulisboa.pt; 3Institute of Clinical Medicine–Neurosurgery, University of Eastern Finland, 70210 Kuopio, Finland; ville.leinonen@kuh.fi; 4Neurology of Neuro Center Kuopio University Hospital, 70210 Kuopio, Finland; 5Research Institute, Shiga Medical Center, Shiga 524-8524, Japan; tanigaki@res.med.shiga-pref.jp; 6Department of Pharmacology, Department of Integrative and Systems Physiology, Department of Cell Systems and Anatomy, Department of Psychiatry, University of Texas Health Science Center, San Antonio, TX 77030, USA; kangg@uthscsa.edu (G.K.); hiroi@uthscsa.edu (N.H.); 7Institute of Clinical Medicine–Neurology, University of Eastern Finland, 70210 Kuopio, Finland; hilkka.soininen@kuh.fi; 8A.I Virtanen Institute for Molecular Sciences, University of Eastern Finland, 70210 Kuopio, Finland

**Keywords:** Aβ, Alzheimer’s disease, APP C-terminal fragments, autophagy, SEPTIN5

## Abstract

Alzheimer’s disease (AD) is a neurodegenerative disease characterized by aberrant amyloid-β (Aβ) and hyperphosphorylated tau aggregation. We have previously investigated the involvement of SEPTIN family members in AD-related cellular processes and discovered a role for SEPTIN8 in the sorting and accumulation of β-secretase. Here, we elucidated the potential role of SEPTIN5, an interaction partner of SEPTIN8, in the cellular processes relevant for AD, including amyloid precursor protein (APP) processing and the generation of Aβ. The in vitro and in vivo studies both revealed that the downregulation of SEPTIN5 reduced the levels of APP C-terminal fragments (APP CTFs) and Aβ in neuronal cells and in the cortex of Septin5 knockout mice. Mechanistic elucidation revealed that the downregulation of SEPTIN5 increased the degradation of APP CTFs, without affecting the secretory pathway-related trafficking or the endocytosis of APP. Furthermore, we found that the APP CTFs were degraded, to a large extent, via the autophagosomal pathway and that the downregulation of SEPTIN5 enhanced autophagosomal activity in neuronal cells as indicated by altered levels of key autophagosomal markers. Collectively, our data suggest that the downregulation of SEPTIN5 increases the autophagy-mediated degradation of APP CTFs, leading to reduced levels of Aβ in neuronal cells.

## 1. Introduction

The main theoretical concept in Alzheimer’s disease (AD) is the amyloid cascade hypothesis, postulating that amyloid-β (Aβ) deposition in the brain is a primary driver of AD pathogenesis, impairing synaptic efficacy, and causes calcium dyshomeostasis, inflammation, oxidative stress, as well as tau hyperphosphorylation and the formation of neurofibrillary tangles (NFTs) at specific brain regions [1]. On the one hand, Aβ is derived from the amyloid precursor protein (APP) by β-amyloidogenic processing via consecutive cleavages by β- and γ-secretases. The initial cleavage by β-secretase (BACE1) sheds an ectodomain of APP (sAPPβ). The remaining membrane-tethered APP carboxy-terminal fragment (C99) is subsequently cleaved by γ-secretase to produce Aβ peptides [2]. On the other hand, the majority of APP is processed via the non-amyloidogenic route by α-secretases, which cleave APP within the Aβ domain, thereby generating sAPPα and C83 fragment, and preventing Aβ generation. The concept of Aβ as a key pathogenic driver in AD has recently been reinforced by the discovery of a protective mutation in *APP* (A673T), which, on the one hand, significantly reduces Aβ production and protects against cognitive decline [3,4]. On the other hand, the causative and fully penetrant genetic mutations in *APP* and *PSEN1/2*, results in enhanced Aβ accumulation, and leads to AD [5]. Nanomolar concentrations of soluble Aβ peptides extracted from the brain of AD patients have been shown to enhance tau pathology in mouse hippocampal neurons, and subsequently to compromise synaptic structure and function, suggesting a synergistic effect of Aβ and tau in neurodegeneration [6]. Thus, identification of factors involved in the early steps of AD pathogenesis, preceding β-amyloid plaque deposition, could possess potential as early biomarkers or therapeutic targets to suppress the development and progression of AD.

SEPTINs are a conserved family of 13 GTP-binding proteins highly expressed in the brain and are involved in processes, such as the regulation of formation, growth, and stability of axons and dendrites; synaptic plasticity; and autophagy, all of which have been identified to be affected during AD pathogenesis [7,8]. Furthermore, we have previously discovered that one of the septin family members, SEPTIN8, modulated Aβ production by affecting the sorting and accumulation of BACE1 [9]. A direct interaction partner of SEPTIN8 is SEPTIN5, which has been identified as an essential component in the regulation of presynaptic vesicle exocytosis [10]. In AD brain, SEPTIN5 has been found present in tau-based paired helical filament cores, suggesting that it could contribute to the formation of neurofibrillary tangles [7,11]. In addition to AD, previous studies have linked SEPTIN5 to Parkinson’s disease (PD), demonstrating that it accumulated in dopaminergic neurons due to parkin dysfunction, resulting in neurotoxicity. SEPTIN5 is ubiquitinated by parkin, which is an E3 ubiquitin-protein ligase promoting the proteasomal turnover of SEPTIN5 [12]. Related to this, familial PD mutations in *PRK2* disrupt the ubiquitin-protein ligase function of parkin, and consequently impair the degradation of SEPTIN5 [12]. Our previous studies showed that SEPTIN5 downregulation led to altered APP processing in human embryonic kidney cells by reducing the levels of soluble APP (sAPP) [9]. Thus, given that Aβ-mediated synaptic dysfunction is one of the earliest features in AD [13] and that SEPTIN5 is known to regulate synaptic vesicle exocytosis and intracellular vesicular trafficking [10], it is essential to further elucidate the role of SEPTIN5 in the cellular processes relevant for AD, such as APP processing and the generation of Aβ.

Here, we set the goal to assess the effects of SEPTIN5 downregulation on APP processing and the generation of Aβ in different in vitro and in vivo neuronal models. Downregulation of SEPTIN5 by using RNA interference (RNAi) in different neuronal cells resulted in decreased levels of APP C-terminal fragments (APP CTFs) and Aβ. The same outcome was observed in the cortical brain lysates obtained from homozygous Septin5 knockout mice. Mechanistic elucidations revealed that the downregulation of SEPTIN5 led to a faster degradation of the APP CTFs. Furthermore, the APP CTFs were found to be degraded to a large extent via the autophagosomal pathway and the downregulation of SEPTIN5 enhanced the autophagosomal activity in the neuronal cells. Collectively, our data suggest that the downregulation of SEPTIN5 increases the autophagy-mediated degradation of APP CTFs, leading to reduced levels of Aβ in vitro and in vivo.

## 2. Materials and Methods

### 2.1. Small Interfering RNAs (siRNAs), Lentiviral shRNAs, and Plasmid Constructs

Silencer^®^ Select Pre-designed and Validated siRNA targeted to SEPTIN5 (5′-AGACGGUAGAGAUUCUAAAtt-3′) (Thermo Fisher Scientific, Waltham, MA, USA, siRNA ID s224294) was used for downregulation of SEPTIN5 expression in SH-SY5Y-APP751 cells. Silencer^®^ Negative control #1 siRNA was used as a control in RNA interference experiments (Thermo Fisher Scientific, Waltham, MA, USA, catalog #4390843). MISSION^®^ shRNA plasmid DNA encoding short hairpins targeted the open reading frame of mouse SEPTIN5 mRNA (Sigma-Aldrich, St. Louis, MO, USA, Clone ID: TRCN0000101511). Third-generation self-inactivating lentiviruses were prepared in triple flasks by a calcium phosphate transfection method in 293T cells, as described previously [14], and concentrated by ultracentrifugation. MISSION^®^ lentiviral control short hairpin transduction particles (Sigma-Aldrich, St. Louis, MO, USA, Clone ID: SHC002H), were used as a control. Plasmid encoding microtubule-associated protein 1B-light chain 3-GFP (GFP-LC3) was used in immunofluorescence studies.

### 2.2. Cell Cultures, Transfections, and Transductions

The human neuroblastoma SH-SY5Y cell line stably overexpressing human APP751 isoform (SH-SY5Y-APP751) was cultured in Dulbecco’s modified Eagle’s medium supplemented with 10% fetal bovine serum (FBS), 2 mM l-glutamine, 100 unit/mL penicillin, 100 µg/mL streptomycin, and 200 µg/mL geneticin. Cells were transfected with 5 nM of SEPTIN5 target or a scrambled control siRNA, and/or 0.8 µg of GFP-LC3 plasmid using Lipofectamine 2000 transfection reagent (Thermo Fisher Scientific, Waltham, MA, USA). The GFP-LC3 construct produces a fusion protein consisting of GFP and LC3 (a kind gift from Dr. Kai Kaarniranta, Department of Ophthalmology, Institute of Clinical Medicine, University of Eastern Finland, Kuopio). Small interfering RNA particle or plasmid DNA-containing media was removed 24 h after transfection. Cells were incubated for another 48 h before collection and analyses. Primary cortical neuronal cultures were dissected and cultured from embryonic day 18 mouse brains (JAXC57BL/6J), as previously described [15]. Cortical neuron cultures were transduced with lentiviral particles at 10–20 multiplicity of infection (MOI) at 5 DIV. The fresh medium was changed 24 h after transductions to wash away excess virus, thereafter cultures were maintained until 13 DIV.

### 2.3. Animals

The generation of Septin5 knockout mice, was previously described [16]. The congenic mouse line was developed by backcrossing the original Septin5 knockout mouse to C57BL/6J mice for more than 10 generations (Harper et al., 2012). This congenic mouse was subsequently backcrossed to C57BL/6N mice for more than 10 generations. Congenic Septin5 homozygous (Septin5^−/−^), heterozygous (Septin5^+/−^) and wild-type (Septin5^+/+^) littermates produced from breeder pairs of heterozygous mice were used. Mouse colonies were maintained in accordance with the protocols approved by the Committee on Animal Research at Research Institute, Shiga Medical Center, Japan (license ID: 27-3). For dissection of tissue samples, three coronal cuts were made using a blade approximately at bregma 2.00 mm, 0.00 mm, and −5.00 mm. The frontal cortex was collected after the first cut. The motor cortex and the striatum were collected after the second cut. The hippocampus and the posterior cortex were collected after the third cut by removing the midbrain. The posterior cortex was cut in the midline to generate the dorsal and temporal cortex. The temporal cortex was used for Western blot and ELISA analysis.

### 2.4. Real-Time Quantitative Polymerase Chain Reaction (PCR) Analysis

RNA extraction was performed using High Pure RNA Isolation kit (Roche, Basel, Switzerland). cDNA was synthesized from the extracted RNA using a Transcriptor First Strand cDNA Synthesis kit (Roche, Basel, Switzerland). Target specific primers for human SEPTIN5 (5′-ACAAGCAGTACGTGGGCTTC-3′ and 5′-TCAGCACTGAGCAGCTTCC-3′), mouse SEPTIN5 (5′-ACGCGGTGAACAACTCTGAA-3′ and 5′-CAGCTTCCGGATCTCACTGG-3′), and GAPDH (5-GGTCTCCTCTGACTTCAACA-3′ and 5′-GTGAGGGTCTCTCTCTTCCT-3′) were designed to amplify the region flanking at least two different exons of the target gene. For quantitative qPCR reactions, SYBR Green FastStart Universal SYBR Green Master (ROX, Roche, Basel, Switzerland) was used, and run on a real-time quantitative PCR machine 7500 Fast Real Time PCR System (Applied Biosystems, Foster City, CA, USA). Standard curve or comparative Ct-method were used for determining SEPTIN5 mRNA levels, where each sample was normalized to its respective house-keeping gene (GAPDH) levels.

### 2.5. Western Blot Analysis

Total protein was extracted from the SH-SY5Y-APP751 cells and mouse primary cortical neurons by scraping the cultures in transmembrane protein extraction reagent (TPER) buffer (Pierce, Waltham, MA, USA) supplemented with protease and phosphatase inhibitors (Thermo Scientific, Waltham, MA, USA). Protein concentrations were determined with bicinchoninic acid assay (Pierce, Waltham, MA, USA). Then, 10–50 µg of total protein lysates were loaded on 4–12% Bis-Tris polyacrylamide gel electrophoresis (Invitrogen, Carlsbad, CA, USA) for protein separation, followed by transfer to a Hybond-P polyvinylidene fluoride (PVDF) membrane (GE Healthcare, Chicago, IL, USA). Nonspecific antibody binding was prevented by incubating the PVDF membranes in 5% non-fat milk (in 1× Tris-buffered saline (TBST)) for 1 h at room temperature (RT). For immunodetection, PVDF membranes were incubated with appropriately diluted primary antibody overnight at 4 °C. The next day, membranes were incubated for 1 h, at RT, with horseradish peroxidase (HRP)-conjugated secondary anti-mouse or anti-rabbit antibodies. Enhanced-chemiluminescence substrate (ECL, GE Healthcare, Chicago, IL, USA) was dispersed on top of the membrane, followed by protein band detection with Syngene Gbox or Chemidoc MP imaging system. Protein band intensities were quantified using Quantity One 1-D Analysis (version 4.6.8) or Image Lab (version 6.1) (Bio-Rad, Hercules, CA, USA) software.

### 2.6. Antibodies

For SEPTIN5 detection, anti-SEPTIN5 rabbit polyclonal antibody 11631-1-AP (1:1000, ProteinTech Group, Deansgate, Manchester, UK) was used. For BACE1 protein detection, rabbit monoclonal anti-BACE1 D10E5 antibody (1:1000, #5606, Cell Signaling Technology, Danvers, MA, USA) was used. Rabbit polyclonal antibody A8717 detecting the APP C-terminus (1:2000, Sigma, St. Louis, MO, USA) was used for detection of APP immature (APPim), APP mature (APPm), and APP CTFs (C83 and C99). APP N-terminal mouse monoclonal antibody MAB348 (1:1000, clone 22C11, Merck Millipore, Burlington, MA, USA) detected total sAPP (sAPPtot). Glyceraldehyde-3-phosphate dehydrogenase (GAPDH) was detected by a mouse monoclonal antibody ab8245 (1:15,000, Abcam, Cambridge, UK). Transferrin receptor was detected with Anti-transferrin receptor mouse monoclonal antibody (1:1000, 13-6800, Invitrogen, Carlsbad, CA, USA). LC3I and LC3II were recognized by anti-LC3B rabbit polyclonal antibody (1:3000, ab51520, Abcam, Cambridge, UK); p62 protein was detected with rabbit polyclonal antibody ab76340 (1:1000, Abcam, Cambridge, UK); and β-actin was detected by mouse monoclonal antibody ab8226 (1:1000, Abcam, Cambridge, UK).

### 2.7. Soluble APP and Aβ Measurements

Soluble sAPPα levels were determined using mouse monoclonal antibody 6E10 (Biosite, Täby, Sweden). sAPPβ was detected using a custom rat monoclonal antibody (BAWT), which binds directly upstream of the β-cleavage site (ISEVKM) (provided by Prof. Stefan Lichtenthaler, DZNE, Munich, Germany, https://www.alzforum.org/antibodies/app-bawt). Total sAPP levels were detected using the mouse monoclonal antibody MAB348 (clone 22C11, Merck Millipore, Burlington, MA, USA), binding to the N-terminus of APP. Aβ40 and Aβ42 levels were quantified from the cell culture media (SH-SY5Y-APP751 and wild-type mouse primary cortical neuron) using Human/Rat β-amyloid 40 (294-64701) (Wako, FUJIFILM Wako Chemicals, VA, USA) and β-amyloid 42 (High-Sensitive, 292-64501) ELISA kits (Wako, FUJIFILM Wako Chemicals, VA, USA), according to manufacturer’s protocol.

### 2.8. Biotinylation of Cell Surface Proteins

First, SH-SY5Y-APP751 cells were washed twice with PBS containing 0.01 mM CaCl_2_ and 1 mM MgCl_2_ (PBS-Ca-Mg), followed by a 15 min preincubation in PBS-Ca-Mg at 4 °C. Sulfo-NHS-LC-Biotin (EZ Link™, Pierce, Waltham, MA, USA) was dispensed onto the cells and incubated 30 min on a rocker at 4 °C. Excess biotin was quenched with 0.1 mM glycine-supplemented PBS-Ca-Mg for 20 min. Subsequently, cells were washed twice with PBS-Ca-Mg and collected in protease inhibitor-supplemented T-PER. Collected cells were centrifuged at 10,000× *g* for 10 min at 4 °C. The supernatant was collected, and protein concentration was determined. Then, 450 µg of proteins were mixed in PBS containing 1% Nonidet P40 and incubated overnight with streptavidin cross-linked agarose beads (Pierce, Waltham, MA, USA). The next day, samples were centrifuged at 5000× *g* for 1 min to collect the biotinylated proteins (pellet) and supernatant containing the unbiotinylated proteins for Western blot analysis.

### 2.9. Cycloheximide Time Course

Cells in a 6-well-plate well were treated with 30 µg/mL cycloheximide and incubated for 0, 20, 40, 60 and 80 min. Thirty µg of total protein from each biological replicate was subjected to Western blot. Protein levels at 20, 40, 60 and 80 min were normalized to protein levels at 0 min and plotted against the time of cycloheximide treatment. To determine whether any change in protein degradation rate over time was a result of downregulation of SEPTIN5, two-way repeated measures ANOVA was used (dependent variable, normalized protein levels; independent variables, time and RNAi treatment).

### 2.10. Immunofluorescence Microscopy and Quantitative Protein Colocalization Analysis

Cells were cultured on coverslips, and fixed in 4% paraformaldehyde (PFA) at RT for 15 min, and permeabilized with 0.1% Triton X-100 for 10 min. To avoid nonspecific antibody binding, coverslips were incubated in blocking solution containing 1.5% (*w*/*v*) goat IgG (Invitrogen, Carlsbad, CA, USA) at RT for 30 min. Subsequently, coverslips were incubated with the following primary antibodies for 1.5 h, at RT: polyclonal antibody detecting SEPTIN5 (1:200, 11631-1-AP, ProteinTech, Deansgate, Manchester, UK) and polyclonal antibody detecting APP C-terminus (1:500, A8717, Sigma-Aldrich, St. Louis, MO, USA). Next, 1 h incubation at RT with secondary Alexa Fluor 594 goat anti-rabbit antibody (1:500, Molecular Probes) was performed. Nuclear staining was accomplished with 4′,6-diamidino-2-phenylindole, dihydrochloride (DAPI). Coverslips were mounted to slides with Fluoromount (F4680, Sigma-Aldrich, St. Louis, MO, USA). Cell imaging was performed using a Zeiss Axio Observer.Z1 inverted microscope (63× NA 1.4 oil objective) equipped with a Zeiss LSM 800 confocal module with Airyscan (Carl Zeiss Microimaging GmbH, Jena, Germany). Images were prepared using Zeiss ZEN 2 core program. Colocalization of APP CTFs and GFP-LC3 in control or SEPTIN5 siRNA-transfected cells was quantified by assessing the number of colocalizing pixels using Zeiss ZEN 2 core colocalization analysis, according to manufacturers’ protocol (https://www.zeiss.com/content/dam/Microscopy/Downloads/Pdf/FAQs/zen-im_colocalization.pdf). Briefly, a region of interest (ROI) was manually drawn to include GFP-LC3-positive cells in the analysis. Every pixel within the ROI was plotted in a scatter diagram based on its intensity level from each channel. From the scatter diagram, the number of colocalizing pixels was quantified, and a weighted colocalization coefficient was calculated using the ZEN Module colocalization analysis tool. Weighted colocalization coefficients were converted to percentages and were shown as a % of colocalizing pixels ± standard error of mean (SEM).

### 2.11. Sample Preparation from SEPTIN5 Knockout Mice

To extract RNA and protein from the temporal cortical tissue sample of 5-month old Septin5 knockout mice, the tissue sample was homogenized in 10× tissue weight of PBS. One-eighth of the homogenized sample was transferred to a separate tube and immersed in 500 µL TRI Reagent^®^ (Molecular Research Center Inc., Cincinnati, OH, USA) for RNA extraction. RNA was extracted with Direct-zol RNA MiniPrep (Zymo Research, Irvine, CA, USA), according to the manufacturers’ protocol. Concentrations and quality of RNA were measured by Nano-Drop ND-2000. SEPTIN5 RNA expression levels were measured by qPCR analysis. The remaining tissue homogenate was centrifuged 100,000× *g* for 1 h, at RT, from where the supernatant was utilized for Aβ42, sAPPtot, and sAPPα quantification by ELISA and Western blot, respectively. The remaining pellet was immersed in 80 µL T-PER, vortexed for 3 h, at RT, and centrifuged 10,000× *g* for 10 min at RT. The remaining supernatant was utilized for Western blot analysis of specific proteins indicated in the manuscript.

### 2.12. Neuron-BV2 Microglia Co-Cultures and Treatments

Primary cortical neuron-BV2 microglia co-cultures were prepared, as described before [15,17]. Briefly, BV2 cells were cultured in RPMI 1640 medium (Sigma-Aldrich, St. Louis, MO, USA) containing 10% FBS, 2 mM L-glutamine, 100 U/mL penicillin, and 100 μg/mL streptomycin. Prior to harvesting, RPMI medium was replaced with Neurobasal medium, whereafter BV2 cells were gently detached with a cell scraper. BV2 cells were seeded on top of DIV11 primary cortical neuron cultures at a 1:5 ratio (BV2 microglia/neurons), and let settle down for 2 h. Neuroinflammation was induced by treating co-cultures with 200 ng/mL of lipopolysaccharide (LPS, Sigma-Aldrich, St. Louis, MO, USA) and 5 ng/mL of interferon-γ (IFNγ, Sigma-Aldrich, St. Louis, MO, USA) for 48 h.

### 2.13. Neuronal Viability Assay and TNF-α Assay

Primary cortical neuron-BV2 co-cultures were fixed by incubating the cells in 4% PFA for 20 min at RT. Subsequently, cells were permeabilized in methanol containing 0.3% H_2_O_2_ for 10 min. Prior to MAP2 staining, nonspecific binding was inhibited by incubation with PBS containing 1% BSA and 10% horse serum (Vector Labs, Orton Southgate, Peterborough, UK) for 20 min, at RT. Neuronal cells were immunolabeled with anti-MAP2 antibody (1:2000, Sigma-Aldrich, St. Louis, MO, USA, M9942) overnight at 4 °C. Next, cells were incubated with biotinylated horse anti-mouse secondary antibody (1:200, Vector Labs, Orton Southgate, Peterborough, UK) for 1 h. ExtrAvidin-HRP tertiary antibody staining (1:500, Sigma-Aldrich, St. Louis, MO, USA) was performed for 1 h, at RT. To develop the staining, cells were incubated with the ABTS peroxidase substrate (Vector Labs, Orton Southgate, Peterborough, UK) for 30 min. From each well, 150 μL of substrate solution was collected and absorbances was measured at 405 nm with a microplate reader (BioRad, Hercules, CA, USA). Levels of secreted TNF-α were determined from 1:2 diluted conditioned culture media with a Ready-SET-Go mouse TNF-α kit (eBioscience, San Diego, CA, USA).

### 2.14. Statistical Analyses

IBM SPSS version 21 was used to analyze the data. Statistical comparisons for two-group comparisons were performed by independent sample *t*-test. For three or more group comparisons, one-way ANOVA followed by Tukey honest significant difference (HSD) or Fisher’s least significant difference (LSD) post hoc test was performed. Statistical analysis of cycloheximide time-course assay was performed using two-way repeated-measures ANOVA followed by Tukey HSD. Results are expressed as mean % ± SEM of control samples. *p*-values < 0.05 were considered to be statistically significant.

## 3. Results

### 3.1. Downregulation of SEPTIN5 Decreases the Levels of Amyloid Precursor Protein (APP) C-Terminal Fragments and Aβ in Neuronal Cells and Septin5 Knockout Mice

We have previously shown that SEPTIN5 downregulation alters APP processing in human embryonic kidney cells by reducing the levels of sAPP fragments using an assay detecting the total levels of sAPP (sAPPtot = sAPPα + sAPPβ) [9]. To elucidate the effects of the downregulation of SEPTIN5 on APP processing and generation of Aβ in neuronal cells, siRNA targeted against SEPTIN5 was applied in the human SH-SY5Y neuroblastoma cells stably overexpressing the APP751 isoform (SH-SY5Y-APP751). siRNA-mediated downregulation of SEPTIN5 resulted in ~50% decrease in the endogenous mRNA and protein levels of SEPTIN5 as compared with the control siRNA-transfected cells (Figure 1A). Downregulation of SEPTIN5 resulted in a significant reduction in the levels of APP CTFs (= APP C83 + C99), accompanied with a significant decrease in the levels of specifically sAPPβ, as well as Aβ40 and Aβ42 in the culture medium (Figure 1A,B). The levels of sAPPtot or sAPPα remained unchanged, suggesting that the observed effects were not related to altered α-secretase-mediated cleavage of APP. Likewise, the fact that the levels of sAPPβ and Aβ were both decreased to a similar extent suggested that γ-secretase-mediated cleavage of APP was not affected owing to the downregulation of SEPTIN5. Additionally, the ratio of Aβ42/Aβ40 remained unchanged, indicating that modulation of γ-secretase activity did not take place. The reduction in the levels of APP CTFs, sAPPβ, Aβ40, and Aβ42 was not associated with altered levels of total full-length APP (APPtot = mature + immature APP) or BACE1 after the downregulation of SEPTIN5 (Figure 1A). Furthermore, the ratio of mature vs. immature APP (APPm/im) was unaltered.

A similar Aβ-related phenotype was observed in the primary cortical neuron cultures prepared from E18 mouse embryos, where both Aβ40 and Aβ42 levels were significantly reduced upon lentiviral shRNA-mediated downregulation of Septin5 (Figure 2A,B). The levels of BACE1, sAPPtot, and sAPPα were not significantly affected in the mouse primary cortical neurons upon the downregulation of Septin5 (Figure 2A,B). However, as with the SH-SY5Y-APP751 cells, a trend towards decreased levels of APP CTFs and sAPPβ was observed. Conversely, the ratio of APPm/im and the levels of endogenous APPtot was significantly decreased upon Septin5 downregulation in primary cortical neurons (Figure 2A). The downregulation of Septin5 did not affect the survival of mouse cortical neurons upon steady state or LPS/IFNγ-induced stress conditions in neuron-BV2 microglia co-cultures, indicating that the changes in APP processing or Aβ generation were not due to decreased neuronal viability (Figure A1).

To elucidate the long-term and dose-dependent effects of Septin5 downregulation on APP processing and generation of Aβ in vivo, we utilized the previously established Septin5 homozygous knockout (Septin5^−/−^), Septin5 heterozygous knockout (Septin5^+/−^), and wild-type (Septin5^+/+^) mice [16,18]. APP processing and the levels of Aβ were analyzed in total protein lysates from the temporal cortex of five-month-old Septin5^+/+^, Septin5^+/−^, and Septin5^−/−^ mice. The mRNA and protein levels of Septin5 were decreased by ~40% in the Septin5^+/−^ mice as compared with the Septin5^+/+^ mice, whereas they were completely undetectable in the Septin5^−/−^ mice (Figure 2C). The levels of APP CTFs were significantly decreased in Septin5^−/−^ mice but not in Septin5^+/−^ mice as compared with the Septin5^+/+^ mice. In parallel, BACE1 protein levels were significantly decreased in Septin5^−/−^ mice as compared with the Septin5^+/+^ mice. Additionally, we observed a significant decrease in Aβ42 levels in Septin5^−/−^ mice, without changes in the levels of sAPPtot or sAPPα (Figure 2D). The levels of sAPPβ and Aβ40 were below detection limits in these samples. We have previously shown that the downregulation of SEPTIN8 increased the degradation of BACE1 [9]. Thus, we next asked whether altered Septin5 expression could result in an alteration in Septin8 levels, which could potentially indirectly explain the observed reduction in BACE1 levels in vivo in Septin^−/−^ ice. Indeed, we observed a dose-dependent decrease in Septin8 levels in the temporal cortex of the Septin5^+/−^ and Septin5^−/−^ mice as compared with the Septin5^+/+^ mice (Figure A2). In conclusion, these in vitro and in vivo data suggest that the downregulation or knock out of SEPTIN5/Septin5 affects APP processing, particularly by reducing the levels of APP CTFs and Aβ in neuronal cells and cortical brain tissue.

### 3.2. Downregulation of SEPTIN5 Increases the Degradation of APP CTFs in Neuronal Cells

Given the above-mentioned APP- and Aβ-related alterations in neuronal cells and brain tissue, we next wanted to delineate the underlying molecular mechanisms in a stepwise manner, by starting from the assessment of APP trafficking onto the cell surface through the secretory pathway. To do this, we performed a cell-surface biotinylation assay in SH-SY5Y-APP751 cells transfected with SEPTIN5 or control siRNA. In the total protein lysates used for the biotinylation assay, the downregulation of SEPTIN5 was confirmed to result in a significant reduction in the levels of APP CTFs, without affecting the levels of APPtot or the ratio of APPm/im (Figure 3A). After pulldown of the biotinylated proteins, SEPTIN5 siRNA and control siRNA-transfected samples showed similar levels of biotinylated mature APP at the cell surface, indicating that the downregulation of SEPTIN5 did not affect the trafficking of APP onto the cell surface (Figure 3B). Moreover, it has been previously shown that inhibition of APP endocytosis in cells expressing APP with mutations in the internalization signal motif “YENP” increased the secretion of sAPPα and reduced the production of Aβ [19]. Hence, the fact that we did not observe changes in the levels of sAPPα in any of the neuronal models studied suggests that the endocytosis of APP remains unaffected after the downregulation of SEPTIN5/Septin5 in neuronal cells or cortical brain tissue. Furthermore, the levels of transferrin receptor (TfR), which is the prototype marker undergoing endocytic recycling [20], were not significantly altered in the biotinylated or total protein fractions upon the downregulation of SEPTIN5 in SH-SY5Y-APP751 cells (Figure 3A,B). This observation suggests that the downregulation of SEPTIN5 does not affect overall endocytosis from the plasma membrane to endosomes.

Since no changes in the secretory pathway-related trafficking or endocytosis of APP were observed, we next assessed whether the downregulation of SEPTIN5 affects the degradation of APP CTFs or APP holoprotein in a cycloheximide time-course experiment in SH-SY5Y-APP751 cells. The GAPDH-normalized levels of APPtot and APP CTFs were investigated at 0, 20, 40, 60 and 80 min after the start of cycloheximide treatment, similar to our previous studies [9,21,22]. Already at 0 min, SEPTIN5 and APP CTF levels were decreased by ~50% in SEPTIN5 siRNA-transfected cells, whereas the levels of APPtot were not changed (Figure 3C). Blocking the de novo protein synthesis by cycloheximide and the subsequent follow-up of protein degradation for up to 80 min in SEPTIN5 siRNA-transfected cells indicated a significantly increased degradation of APP CTFs, but not APPtot, as compared with the control cells (Figure 3D). The levels of APP CTFs remained unaltered and were close to 100% in siControl siRNA-transfected cells throughout the 80 min time course. This observation is potentially related to the fact that APP CTFs are primarily degraded via the autophagosomal-lysosomal degradation pathway, and that cycloheximide, in addition to blocking de novo protein synthesis, also moderately inhibits the autophagosomal-lysosomal pathway [23,24]. Taken together, these data suggest that the downregulation of SEPTIN5 increases the degradation of APP CTFs, without affecting the secretory pathway-related trafficking or the endocytosis of APP in SH-SY5Y-APP751 cells.

### 3.3. APP CTFs Are Degraded via the Autophagosomal Pathway and the Downregulation of SEPTIN5 Enhances the Autophagosomal Activity in Neuronal Cells

The autophagosomal-lysosomal degradation pathway has previously been described to be responsible for the removal of APP CTFs and Aβ [23]. More specifically, it has been shown that treating neuronal N2a cells with an autophagy inducer led to decreased levels of APP CTFs and Aβ, which was reversed by treatment with the autophagy inhibitor bafilomycin A1 (BfA1). BfA1 inhibits the fusion of autophagosomes, which carry proteins destined for degradation, with lysosomes. As the downregulation of SEPTIN5 increased the degradation of APP CTFs, we first wanted to confirm that APP CTFs were degraded via the autophagosomal pathway also in the SH-SY5Y-APP751 cells. Treatment of SH-SY5Y-APP751 cells with BfA1 for 2 h led to an average two-fold increase in the levels of APP CTFs alongside with an increased ratio of LC3II vs. LC3I (LC3II/LC3I), which is a well-established index of autophagosomal accumulation (Figure 4A). Serum-starved SH-SY5Y-APP751 cells were used as controls to detect the correct LC3I and LC3II bands from the Western blot gels. Since the levels of APPtot remained unchanged upon BfA1 treatment, these data indicated that specifically the APP CTFs were degraded to a significant extent via the autophagosomal pathway in SH-SY5Y-APP751 cells. Although the levels of APP CTFs were significantly decreased in the untreated samples upon the downregulation of SEPTIN5 as compared with the untreated control samples, the difference between BfAI-treated siSEPTIN5 and siControl samples with respect to APP CTFs was not significant. Analysis of the key autophagy markers LC3I, LC3II, and p62 in both untreated and BfAI-treated samples revealed that the downregulation of SEPTIN5 significantly decreased the levels of LC3I, LC3II, and p62 as compared with the control samples (Figure 4A). Similarly, a significant decrease in LC3I and LC3II protein levels, accompanied by a slight decrease in p62 levels, was observed in mouse primary cortical neurons upon the downregulation of Septin5 (Figure 4B). LC3 protein levels also showed a trend towards a dose-dependent decrease in Septin5^+/−^ and Septin5 ^−/−^ mice as compared with Septin5^+/+^ mice, however, the decrease did not reach statistical significance. The p62 levels remained unaltered (Figure 4B). However, the downregulation of SEPTIN5 did not significantly alter the ratio of LC3II/I in SH-SY5Y-APP751 cells or mouse primary cortical neurons as compared with the control cells, suggesting that the conversion rate of LC3I to LC3II was not affected. To delineate whether the observed decrease in the levels of LC3 upon downregulation of SEPTIN5 was related to changes in expression or degradation, the de novo protein synthesis was blocked by cycloheximide during an 80 min time course in SH-SY5Y-APP751 cells (Figure 4C). LC3I was degraded significantly faster during the 80 min time course in SEPTIN5 siRNA-transfected cells as compared with the control siRNA-transfected cells (Figure 4C). This suggests that the degradation rather than the expression of LC3 is enhanced due to the downregulation of SEPTIN5 in SH-SY5Y-APP751 cells. Thus, in relation to the previous data on investigating alterations in LC3I, LC3II, and p62 levels upon autophagy stimulation or inhibition [25], our data suggest that the downregulation of SEPTIN5 enhances autophagosomal activity in the neuronal cells.

### 3.4. SEPTIN5 Colocalizes with GFP-LC3-Positive Autophagosomal Vesicles and the Downregulation of SEPTIN5 does not Alter the Recruitment Dynamics of APP CTFs to Autophagosomal Vesicles

As our data suggest a role for SEPTIN5 in the regulation of autophagy, we next asked whether SEPTIN5 localizes to LC3-positive autophagosomal vesicles in SH-SY5Y-APP751 cells. By transfecting SH-SY5Y-APP751 cells with a GFP-tagged LC3 plasmid and the subsequent immunofluorescence analysis of endogenously expressed SEPTIN5, it was found that SEPTIN5 localizes in the GFP-LC3-positive vesicles (Figure 5A). SEPTIN5 protein was also detected near the cell surface, which was consistent with previous findings [26]. As SEPTIN5 colocalized with GFP-LC3-positive vesicles, we next assessed whether the downregulation of SEPTIN5 affected the recruitment of APP CTFs to GFP-LC3-positive vesicles. The downregulation of SEPTIN5 (Figure 5B) did not alter the proportion of APP CTFs colocalized with GFP-LC3 as compared with the control siRNA-transfected vehicle- or BfA1-treated cells (Figure 5C), suggesting that the downregulation of SEPTIN5 did not alter the recruitment dynamics of APP CTFs to the autophagosomal vesicles. However, in line with the Western blot data in Figure 4A, the downregulation of SEPTIN5 resulted in a decreased proportion of GFP-LC3 colocalized with APP CTFs, which was most likely due to the reduced levels of APP CTFs in these cells (Figure 5C). Importantly, APP CTFs and GFP-LC3 colocalization was significantly increased in BfA1-treated samples as compared with the vehicle-treated cells. Taken together, our immunofluorescence data suggest that SEPTIN5 colocalizes with GFP-LC3-positive autophagosomal vesicles and the downregulation of SEPTIN5 does not alter the recruitment dynamics of APP CTFs to autophagosomal vesicles.

## 4. Discussion

Several SEPTINs have been identified to take part in biological processes that are aberrantly affected in AD pathogenesis, such as autophagy and synaptic plasticity [7]. Furthermore, we recently described that one of the SEPTIN family members, SEPTIN8, modulates Aβ production by affecting the sorting and accumulation of BACE1 [9]. Despite these evident links, studies addressing the potential role of SEPTINs in AD pathogenesis remains sparse. Here, we have now assessed the role of the presynaptic vesicle protein SEPTIN5 in cellular processes relevant for AD with a particular focus on APP processing and the generation of Aβ. The in vitro and in vivo studies both indicated that the downregulation of SEPTIN5 significantly reduced the levels of APP CTFs and Aβ in neuronal cells and in the cortex of homozygous Septin5 knockout mice. In turn, mechanistic investigations revealed that SEPTIN5 colocalized in GFP-LC3 positive vesicles and that the downregulation of SEPTIN5 led to the enhanced degradation of APP CTFs via the autophagosomal pathway. Despite causing increased APP CTF degradation, the downregulation of SEPTIN5 did not alter the recruitment dynamics of APP CTFs to autophagosomal vesicles, suggesting that the increased APP CTF degradation was linked to enhanced autophagosomal activity in SEPTIN5 knockdown cells.

SEPTIN family members, namely SEPTIN2, SEPTIN6, SEPTIN7, and SEPTIN9, have previously been described to mediate autophagy of bacteria [27] by forming cage-like structures around intracytosolic bacteria, and hence enabling their autophagic degradation. Moreover, the downregulation of SEPTIN2 or SEPTIN9 in the presence or absence of an autophagosomal inhibitor resulted in a robust decrease in the levels of LC3 and p62 in HeLa cells, indicating that these critical components of the autophagosomal pathway were markedly affected by the depletion of certain SEPTIN family members [27]. These are important findings given the fact that we observed a similar outcome in terms of LC3 and p62 levels upon the downregulation of SEPTIN5. This suggests that these specific SEPTIN family members play a similar functional role in the regulation of autophagy. In general, autophagy is an essential cellular pathway taking care of the degradation of, for example, misfolded and aggregated proteins, and increasing evidence points to an altered autophagy function in AD, a disease characterized by accumulation of aggregated proteins in the brain [28]. Autophagy has previously been identified as the primary degradation pathway for APP CTFs and Aβ, where the AP2 complex functions as a mediator that bridges APP CTFs to LC3 in autophagosomes during endocytosis [29]. Thus, the interactions among LC3, AP2 complex, and APP CTFs are analogous to that observed in the case of LC3 and p62, which functions as a receptor protein that targets other proteins to selective autophagy and itself undergoes autophagosomal degradation. The fact that induction of autophagy by rapamycin in vivo has been shown to lower intracellular levels of Aβ and improve cognition in an AD mouse model suggests that the modulation of autophagosomal activity could represent a potential therapeutic approach in AD [30].

Our present data suggest, for the first time, a role for SEPTIN5 in the regulation of autophagy. To support this idea, downregulation of SEPTIN5 was found to significantly increase the degradation rate of APP CTFs and Aβ, without affecting the secretory pathway-related trafficking of APP, endocytosis of APP, or the levels of total APP or BACE1 in neuronal cells. Furthermore, the downregulation of SEPTIN5 significantly decreased the protein levels of LC3I, LC3II, and p62. This suggests augmented autophagosomal activity in neuronal cells, which underlies the enhanced degradation of APP CTFs and the subsequent decrease in the levels of Aβ. This notion was further reinforced by the discovery that the degradation of LC3I was enhanced upon the downregulation of SEPTIN5, as indicated by the cycloheximide time course. Similar outcome measures with respect to APP CTFs, Aβ, LC3I, and LC3II were also observed in primary mouse cortical neurons upon the downregulation of SEPTIN5 and in the temporal cortex of homozygous Septin5 knockout mice. Importantly, treatment with BfA1, an inhibitor of autophagosome-lysosome fusion, robustly increased the levels of APP CTFs, reinforcing the concept that APP CTFs were degraded via the autophagosomal pathway in neuronal cells, as also shown previously [29]. Moreover, we confirmed that the effects of SEPTIN5 downregulation on APP CTFs, Aβ, and LC3 did not result from altered neuronal viability in the steady-state or LPS- and IFNγ-induced stress conditions in neuron-BV2 microglia co-cultures. Taken together, our data suggest that the downregulation of SEPTIN5 increases the autophagy-mediated degradation of APP CTFs, leading to reduced levels of Aβ in neuronal cells.

In addition to the effects of SEPTIN5 knockdown on the levels of APP CTFs and Aβ, downregulation of the SEPTIN5 in primary mouse cortical neuron cultures showed significantly decreased total levels of APP and reduced APPm/im ratio. This suggests that SEPTIN5 may additionally affect the trafficking of APP through the secretory pathway in polarized cells, such as neurons, but not in non-polarized neuroblastoma cells. Moreover, homozygous Septin5 knockout mice showed a moderate reduction in the levels of BACE1. The ubiquitin-proteasome pathway has previously been shown to mediate BACE1 degradation, but recent findings have also suggested a role for autophagy in the degradation of BACE1 [31,32]. Thus, the observed reduction in BACE1 levels in Septin5 knockout mice could possibly be explained by enhanced autophagosomal degradation. However, the reduction in BACE1 levels was only detected in homozygous but not in heterozygous Septin5 knockout mice or after the downregulation of SEPTIN5 in neuroblastoma or cultured primary mouse cortical cells, suggesting that a complete loss of Septin5 was needed for this effect. We have previously identified SEPTIN8, another SEPTIN family protein and an interaction partner of SEPTIN5, to modulate intracellular trafficking and degradation of BACE1. Therefore, it is possible that knock out of SEPTIN5 could influence the turnover of BACE1 also through affecting SEPTIN8 function [9,33]. Accordingly, we observed a dose-dependent decrease in Septin8 protein levels in vivo in Septin5^+/−^ and Septin5^−/−^ mouse temporal cortex as compared with Septin5^+/+^ mice. However, modulation SEPTIN5 levels did not have a similar effect on BACE1 levels in SH-SY5Y-APP751 cells or cultured primary cortical neurons, warranting further mechanistic investigation to fully understand the observed effect in Septin5 knockout mice. However, these data support the idea that at least in vivo in mouse brain, the regulation of SEPTIN5 and SEPTIN8 levels is interlinked, and alteration of the levels of either SEPTIN may have an impact on the levels, and subsequently function of the other. Collectively, our findings suggest that, in addition to modulating the degradation of APP CTFs and the levels of Aβ, SEPTIN5 may modulate the secretory pathway-related trafficking of APP in a cell type-specific manner, particularly in polarized neurons.

Since our data suggested that the downregulation of SEPTIN5 increases the autophagy-mediated degradation of APP CTFs, leading to reduced levels of Aβ in the investigated in vitro and in vivo models, we wanted to confirm whether SEPTIN5 colocalized with autophagosomal vesicles or altered the recruitment dynamics of APP CTFs to autophagosomal vesicles. SEPTIN5 was found to colocalize with GFP-LC3-positive autophagosomal vesicles in neuroblastoma cells. However, downregulation of SEPTIN5 did not affect the colocalization of APP CTFs with GFP-LC3. This, together with the data showing that the downregulation of SEPTIN5 decreased the levels of LC3I, LC3II and p62, suggested that SEPTIN5 did not play a role in the induction of autophagy or in the trafficking of the endocytic cargo to the autophagosomes, but rather influenced the later phases of autophagy. Additionally, we found that ~70% of APP CTFs colocalized with GFP-LC3, further supporting the important role of autophagy in the degradation of APP CTFs in the neuroblastoma cells. Given the GTP-binding nature of SEPTINs, and that SEPTINs are known to associate with microtubules, SEPTIN5 might also regulate autophagosome trafficking along the microtubules through interactions with Rab small GTPase(s) [34]. Growing evidence suggests that members of the Rab small GTPase protein family play key roles in the regulation of autophagy, with Rab7 identified as a key modulator of autophagosome transport along microtubules and autophagosome-lysosome fusion [35]. However, further studies are needed to delineate whether SEPTIN5 plays a role in the regulation of autophagosome trafficking along the microtubules.

In conclusion, here, our findings suggest a novel role for the presynaptic vesicle protein SEPTIN5 as a regulator of autophagy similarly to the other SEPTIN family member proteins SEPTIN2 and SEPTIN9 [27]. Importantly, our data indicate that the modulation of SEPTIN5 levels affects the autophagy-mediated degradation of the APP CTFs and the levels of Aβ in neuronal cells both in in vitro and in vivo. Since increasing evidence points to alterations in autophagy in AD, it is pivotal to identify individual autophagy-related factors, which can affect relevant cellular processes in AD pathogenesis, and which subsequently could be used as specific targets for therapeutic interventions in AD.

## Figures and Tables

**Figure 1 cells-09-02482-f001:**
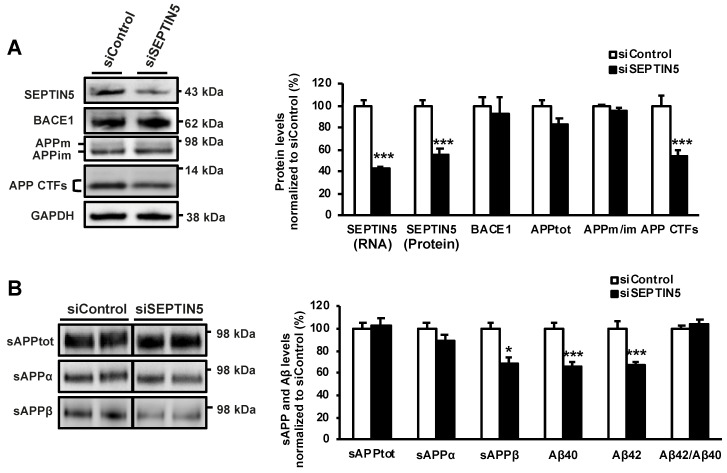
Downregulation of SEPTIN5 decreases the levels of amyloid precursor protein (APP) C-terminal fragments, sAPPβ and Aβ in SH-SY5Y-APP751 cells. (**A**) Quantitative PCR and Western blot analysis to assess the effects of the downregulation of SEPTIN5 (siSEPTIN5) on APP processing in SH-SY5Y-APP751 cells. The levels of APP C-terminal fragments (CTFs) showed a significant decrease upon the downregulation of SEPTIN5 as compared with the control cells (siControl); (**B**) Levels of sAPPβ (Western blot) and Aβ (ELISA) measured from cell culture medium indicated a decrease upon SEPTIN5 downregulation. The levels of sAPPtot and APPα remained unchanged. In (A), protein levels were normalized to GAPDH and are shown as a % of siControl. In (B), protein levels were normalized to total protein levels in the respective cell lysates and are shown as a % of siControl. *n* = 4–0, mean ± SEM, independent *t*-test, * *p* < 0.05, *** *p* < 0.001.

**Figure 2 cells-09-02482-f002:**
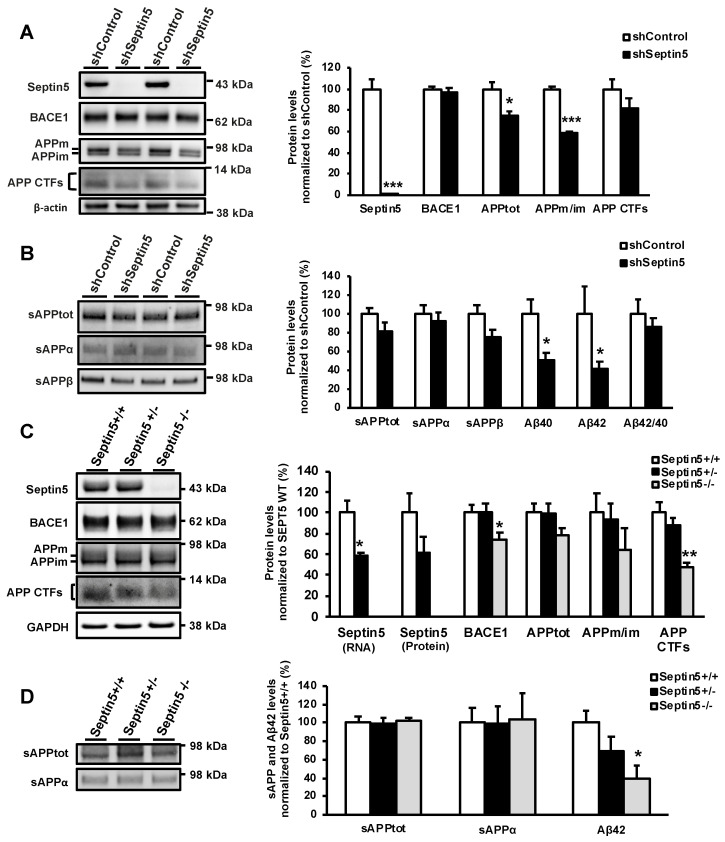
Downregulation of SEPTIN5 affects APP processing and the levels of Aβ in primary mouse cortical neurons and Septin5 knockout mice. (**A**) Western blot analysis showing a significant downregulation of mouse Septin5 and the ratio of APPm/im in mouse primary cortical neurons transduced with lentiviral Septin5 shRNA; (**B**) Aβ40 and Aβ42 levels (ELISA) are significantly decreased in mouse primary cortical neurons after the downregulation of Septin5 as compared with the control-transduced neurons. *n* = 4; (**C**) Western blot analysis of ventral cortical protein lysates from five-month-old Septin5^+/+^, Septin5^+/−^, Septin5^−/−^ mice. The levels of BACE1 and APP CTFs were significantly reduced in Septin5^−/−^ mice. Septin5^+/+^ (*n* = 5), Septin5^+/−^ (*n* = 5), Septin5^−/−^ (*n* = 4); (**D**) Analysis of sAPPtot, sAPPα, and Aβ levels in the temporal cortex of Septin5^+/+^, Septin5^+/−^, Septin5^−/−^ mice. The levels of Aβ (ELISA) were significantly reduced in Septin5^−/−^, but not in Septin5^+/−^ mice, suggesting that the effects of Septin5 downregulation on Aβ were dose dependent. Septin5^+/+^ (*n* = 7), Septin5^+/−^ (*n* = 7), Septin5^−/−^ (*n* = 3). In (A, C) protein levels were normalized to β-actin and GAPDH, respectively, and are shown as a % of siControl/Septin5^+/+^, mean ± SEM. In (B) and (D), protein levels were normalized to total protein levels in the respective cell lysates and are shown as a % of siControl/Septin5^+/+^, mean ± SEM. A/B, independent *t*-test. * *p* < 0.05, ** *p* < 0.01, *** *p* < 0.001. C/D, one-way ANOVA post hoc LSD. * *p* < 0.05 and ** *p* < 0.01.

**Figure 3 cells-09-02482-f003:**
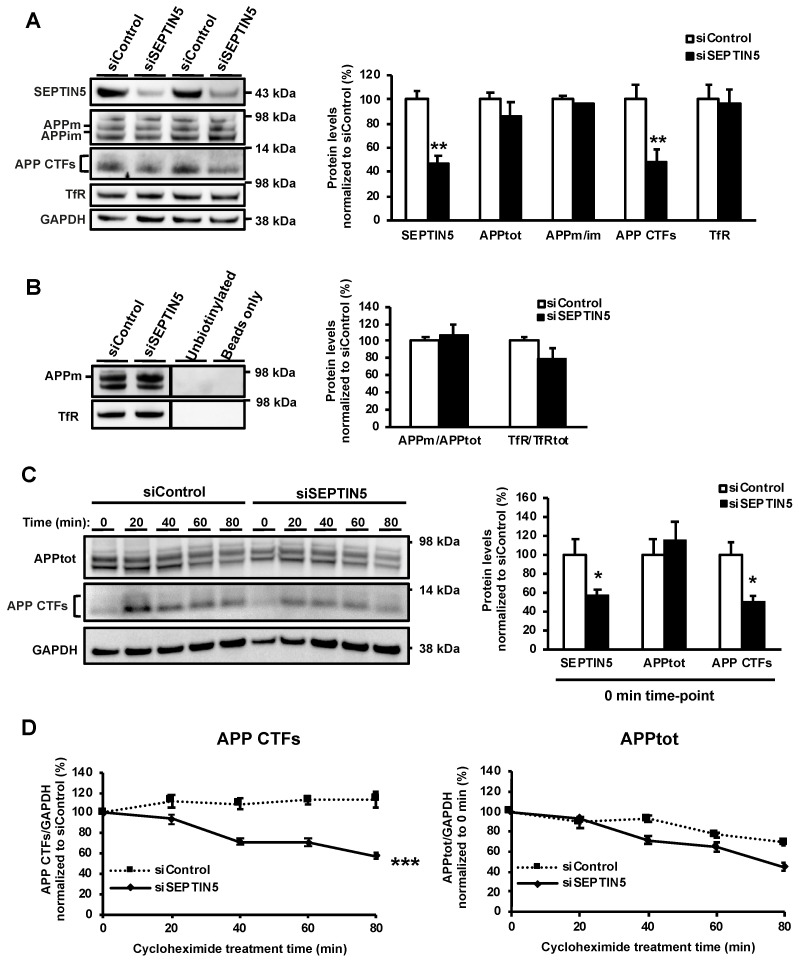
Downregulation of SEPTIN5 does not affect the trafficking of APP to the plasma membrane but increases the degradation of APP CTFs in SH-SY5Y-APP751 cells. (**A**) Western blot analysis of the siSEPTIN5-transfected SH-SY5Y-APP751 cells showed significantly decreased levels of SEPTIN5 and APP CTFs; (**B**) Western blot analysis after cell surface biotinylation showing that downregulation of SEPTIN5 does not affect mature APP (APPm) or transferrin receptor (TfR) levels at the cell surface. Cell surface APPm and TfR levels were normalized to the total levels of APP and TfR in the respective cell lysates and are shown as a % of siControl, *n* = 4, mean ± SEM, independent *t*-test, ** *p* < 0.01; (**C**) Western blot analysis of SH-SY5Y-APP751 cells transfected with SEPTIN5 small interfering RNA (siRNA) and treated with 30 µg/mL cycloheximide for 0, 20, 40, 60, and 80 min. The levels of APP CTFs are already significantly decreased at 0 min upon the downregulation of SEPTIN5 as compared with control siRNA-transfected cells; (**D**) Quantification of the levels of APP CTFs at different times after cycloheximide treatment showed a significant increase in the degradation rate of APP CTFs upon the downregulation of SEPTIN5 as compared with control siRNA-transfected cells. The degradation rate of APPtot is unaffected by the downregulation of SEPTIN5. Protein levels were normalized to GAPDH and are shown as a % of siControl. (**A**) *n* = 4, mean ± SEM, independent *t*-test. (**B**) *n* = 4, SEM, two-way repeated measures ANOVA, post hoc Tukey HSD. * *p* < 0.05, ** *p* < 0.01, *** *p* < 0.001.

**Figure 4 cells-09-02482-f004:**
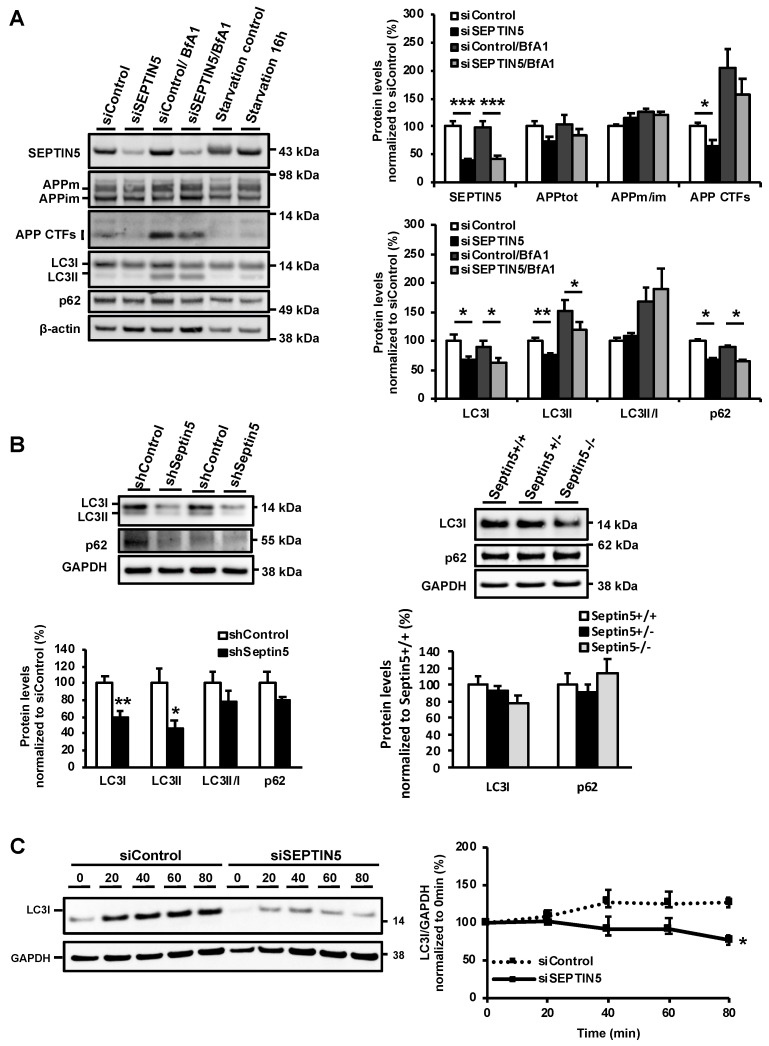
APP CTFs are degraded via the autophagosomal pathway and the downregulation of SEPTIN5 enhances autophagosomal activity in SH-SY5Y-APP751 cells. (**A**) Western blot analysis of SH-SY5Y-APP751 cells transfected with SEPTIN5 siRNA or control siRNA and treated with 200 ng/mL BfA1 or vehicle. Additionally, protein levels from serum-starved (16 h) and starvation control (normal culture medium containing 10% FBS) in SH-SY5Y-APP751 cells were used as controls. Protein levels were normalized to β-actin and are shown as a % of siControl, *n* = 7, mean ± SEM, independent *t*-test, * *p* < 0.05, ** *p* < 0.01, *** *p* < 0.01; (**B**) Western blot analysis of LC3I, LC3II, and p62 in mouse primary cortical neurons transduced with Septin5 shRNA and in Septin5^+/+^, Septin5^+/−^, and Septin5^−/−^ mice. Protein levels were normalized to GAPDH and are shown as a % of siControl or Septin^+^^/+^ mean ± SEM. Independent *t*-test, * *p* < 0.05, ** *p* < 0.01, *** *p* < 0.01 (left figure). One-way ANOVA, post-hoc LSD (right figure); (**C**) Western blot analysis of LC3I in SH-SY5Y-APP751 cells transfected with SEPTIN5 siRNA and treated with 30 µg/mL cycloheximide for 0, 20, 40, 60, and 80 min, *n* = 4, SEM, two-way repeated measures ANOVA, post hoc Tukey HSD. * *p* < 0.05.

**Figure 5 cells-09-02482-f005:**
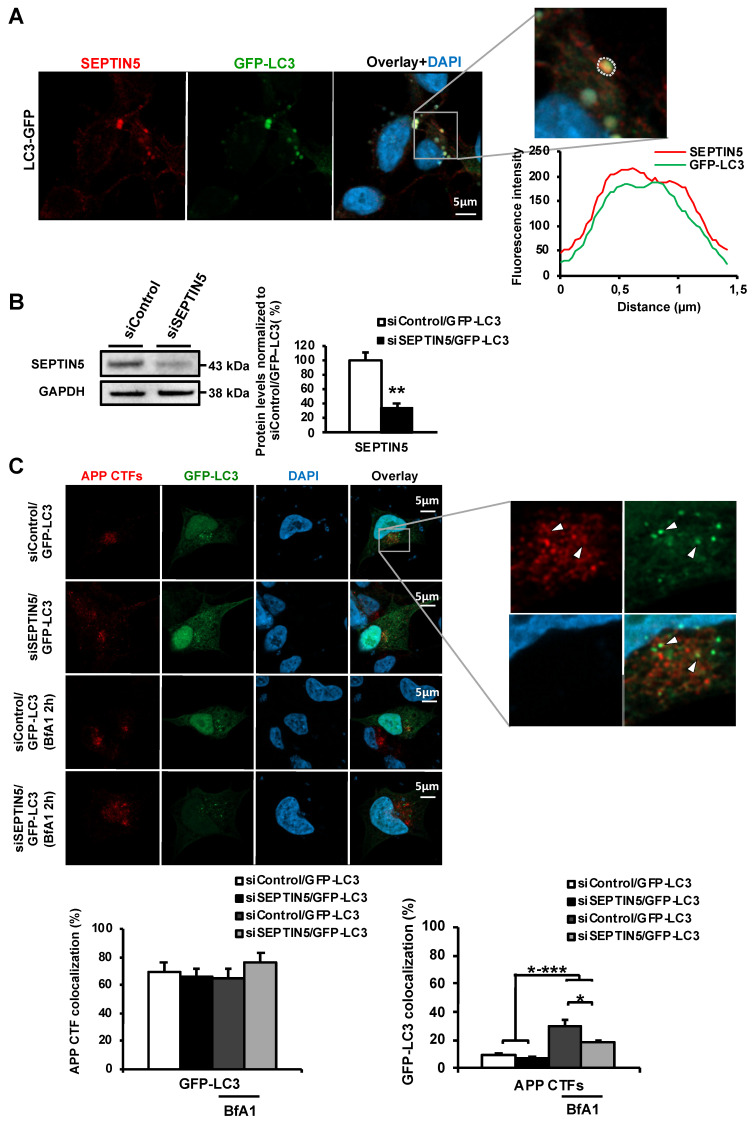
SEPTIN5 colocalizes with GFP-LC3-positive autophagosomal vesicles and the downregulation of SEPTIN5 does not alter the colocalization of APP CTFs with GFP-LC3 in SH-SY5Y-APP751 cells. *(***A**) Superresolution confocal images of endogenous SEPTIN5 (red) in immunostained GFP-LC3-transfected SH-SY5Y-APP751 cells. Quantification of the overlapping fluorescence intensity of a GFP-LC3 positive vesicle shows a marked colocalization for SEPTIN5 and GFP-LC3 (region of interest is indicated with white dotted line); (**B**) Western blot analysis showing SEPTIN5 downregulation in SEPTIN5 siRNA-transfected SH-SY5Y-APP751 cells, *n* = 4, mean ± SEM, independent *t*-test, ** *p* < 0.01; (**C**) Immunostaining of APP CTFs (red) in GFP-LC3 and SEPTIN5 or control siRNA co-transfected SH-SY5Y-APP751 cells, treated with BfA1 (200 ng/mL) or vehicle. Quantitative analysis of APP CTF and GFP-LC3 colocalization showed a significant increase in the proportion of GFP-LC3 colocalized with APP CTFs in BfA1-treated samples as compared with the vehicle-treated cells. The proportion of GFP-LC3 colocalizing with APP CTFs was significantly decreased in cells with SEPTIN5 downregulation and treated with BfA1 as compared with the control siRNA and BfA1-treated cells. siControl/GFP-LC3 *n* = 19, siSEPTIN5/GFP-LC3 *n* = 20, siControl/GFP-LC3/BfA1 *n* = 19, siSEPTIN5/GFP-LC3/BfA1 *n* = 18, mean ± SEM, one-way ANOVA, post hoc Tukey HSD, * *p* < 0.05, ** *p* < 0.01, *** *p* < 0.001.

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
