# Peer review of "Presynaptic Vesicle Protein SEPTIN5 Regulates the Degradation of APP C-Terminal Fragments and the Levels of Aβ"

_cells, 2020, doi:10.3390/cells9112482_

Round 1
Reviewer 1 Report
Marttinen et al report the effect of SEPTIN5 silencing or genetic deletion on APP processing and betaCTF/amyloid levels by resorting to neuroblastoma cell line SH-SY5Y, primary cortical neurons, and SEPTIN5 KO mouse tissues. In absence of SEPTIN5, the amyloidogenic route of APP is affected, without any observe effect on total APP levels or sAPP, suggesting a specific role of SEPTIN5 on APP degradation, possibly via the autophagosomal route.
The manuscript is well written in a standard English, with minor typing errors.
Main Points:
Fig. 1B and graph: why the Abeta 40, 42 WB bands quantified were not shown?
Fig.1 Legend: Please, explain why total protein lysates instead of tubulin III or other loading controls were used here for normalization.
Line 326: please justify the use of “ventral cortex” and describe which part of the cortex was dissected and how in M&M section.
Fig. 2A: the APP immature and mature bands are too tight to be separately analysed and quantified by optical density. Generation of CTFs is dependent upon gamma secretase cleavage. ADAM10/17 level should also be investigated.
Fig.2B and graph: why the Abeta 40, 42 WB bands quantified were not shown?
Fig. 2C: the same as Fig.1A for APPi/APPm bands
Fig. 2D: the same as Fig. 1B for amyloid 42 bands
Fig. 3C: authors argument that total APP level are unchanged following SEPTIN5 siRNA silencing, but the band at 80’ is significantly lower density, as compared to 0-60’ time points. Please, explain, recalculate and/or replace with a more representative picture
Lines 431-433 (Fig. 4B): LC3I level in KO tissues are comparable to wt mice. There is any significant difference in the graph, neither in the WB bands shown.
Overall, the characterization of cellular and molecular mechanisms underlying SEPTINs control of APP is certainly of interest for a deeper understanding of Alzheimer’s Disease etiopathology. However, the effect of SEPTIN5 seams to also extend to BACE1, and potentially to other not investigated targets, arguing its relevance as putative therapeutic target.
Author Response
Reviewer 1
Marttinen et al report the effect of SEPTIN5 silencing or genetic deletion on APP processing and betaCTF/amyloid levels by resorting to neuroblastoma cell line SH-SY5Y, primary cortical neurons, and SEPTIN5 KO mouse tissues. In absence of SEPTIN5, the amyloidogenic route of APP is affected, without any observe effect on total APP levels or sAPP, suggesting a specific role of SEPTIN5 on APP degradation, possibly via the autophagosomal route.
The manuscript is well written in a standard English, with minor typing errors.
- We thank the reviewer for the comments. Below we provide point-by-point answers.
Fig. 1B and graph: why the Abeta 40, 42 WB bands quantified were not shown?
- This is a relevant point, and we apologize for leaving it unclear in the previous version of the manuscript. Ab40 and Ab42 levels were measured using ELISA, and not WB. This has now been clarified in the Figure 1 caption as follows:
“B) Levels of sAPPβ (Western blot) and Aβ (ELISA) measured from cell culture medium indicated a decrease upon SEPTIN5 downregulation. The levels of sAPPtot and APPα remained unchanged.”
Fig.1 Legend: Please, explain why total protein lysates instead of tubulin III or other loading controls were used here for normalization.
- The sAPPtot, sAPPa, sAPPβ, and Ab are secreted to the cell culture media and, thus, these measurements were performed from the media. Only secreted proteins are present in the media, and this does not include classical housekeeping proteins, such as tubulin III. To normalize the data, we used normalization against total protein concentrations as similar number of cells was plated in each culture well at the beginning of the experiment and therefore the total protein amounts, from which the secreted proteins are derived from, are expected to be similar in all the samples. This is a commonly accepted procedure used in the literature [1–3].
Line 326: please justify the use of “ventral cortex” and describe which part of the cortex was dissected and how in M&M section.
- Additional clarification of the dissection procedure has been now provided in detail in the Material and Methods part as stated below. The term ventral cortex has now been accordingly substituted with the term temporal cortex. The temporal cortical samples were used due to the high relevance of the brain region (encompassing e.g. the entorhinal region) in Alzheimer’s disease. The revised text in the manuscript is now as follows:
“For dissection of tissue samples, three coronal cuts were made using a blade approximately at bregma 2.00mm, 0.00mm and -5.00mm. The frontal cortex was collected after the first cut. The motor cortex and the striatum were collected after the second cut. The hippocampus and the posterior cortex were collected after the third cut by removing the midbrain. The posterior cortex was cut in the midline to generate the dorsal and temporal cortex. The ventral posterior cortex was used for Western blot and ELISA analysis.”
Fig. 2A: the APP immature and mature bands are too tight to be separately analysed and quantified by optical density. Generation of CTFs is dependent upon gamma secretase cleavage. ADAM10/17 level should also be investigated.
- We respectfully disagree with the reviewer that APP immature and mature bands in Figure 2A do not have sufficient separation for quantification. Quantification of APPm/im levels via Western blot is a routine and well-established procedure conducted in our research group and consequently, we have applied this protocol in our previously published articles many times [1–3]. The Reviewer is correct that the generation of APP CTF is dependent on γ-secretase cleavage, and possible variation in ADAM10/17 could alter APP processing. However, if our findings were related to altered γ-secretase activity, we should observe an inverse relationship between sAPPβ and Aβ levels. Importantly, we observed here a decrease in both sAPPβ and Aβ levels. Also, the ratio of Aβ42/Aβ40 remained unchanged, indicating that the modulation of γ-secretase activity did not take place. Thus, we can conclude that the downregulation of SEPTIN5 does not affect or modulate γ-secretase activity. Conversely, if the levels of α-secretases (ADAM10 or 17) or activity were affected, the levels of sAPPα should be changed in the culture medium. We did not observe any changes in the levels of sAPPα. However, to address the Reviewer’s point, we have now clarified this point in the text as follows:
Lines 293-298
“The levels of sAPPtot or sAPPα remained unchanged, suggesting that the observed effects are not related to altered α-secretase-mediated cleavage of APP. Likewise, the fact that the levels of sAPPβ and Aβ were both decreased to a similar extent suggests that γ-secretase-mediated cleavage of APP is not affected owing to the downregulation of SEPTIN5. Also, the ratio of Aβ42/Aβ40 remained unchanged, indicating that modulation of γ-secretase did not take place.”
Fig.2B and graph: why the Abeta 40, 42 WB bands quantified were not shown?
- As previously indicated above, Ab40 and Ab42 levels were measured by ELISA and not WB. This has been clarified in the Figure 2B caption as follows:
“B) Aβ40 and Aβ42 levels (ELISA) are significantly decreased in mouse primary cortical neurons after the downregulation of Septin5 as compared to control-transduced neurons. n=4.”
Fig. 2C: the same as Fig.1A for APPi/APPm bands
- Please see the answer for the comment related to Figure 2A.
Fig. 2D: the same as Fig. 1B for amyloid 42 bands
- As explained previously, Ab40 and Ab42 levels were measured by ELISA, not by WB . This has been clarified in the Figure 2D caption as follows:
“The levels of Aβ (ELISA) were significantly reduced in Septin5-/-, but not in Septin5+/- mice, suggesting that the effects of Septin5 downregulation on Aβ were dose-dependent.”
Fig. 3C: authors argument that total APP level are unchanged following SEPTIN5 siRNA silencing, but the band at 80’ is significantly lower density, as compared to 0-60’ time points. Please, explain, recalculate and/or replace with a more representative picture
- We respectfully argue that the downregulation of SEPTIN5 in SH-SY5Y-APP751 cells does not alter the degradation rate of total APP in the steady state situation compared to control cells (Figure 3C). Importantly, we do not state that the downregulation of SEPTIN5 would stabilize total APP and thus, the expected degradation of APP is observed, resulting in a fainter signal at the 80 min timepoint, as compared to 0-60 min timepoints. The observed degradation rate of APP in the present manuscript is in line with the previously published data on APP degradation rate [4]. Moreover, there were no statistically significant changes in the total APP levels between the siControl and siSEPTIN5 at any of the time points.
Lines 431-433 (Fig. 4B): LC3I level in KO tissues are comparable to wt mice. There is any significant difference in the graph, neither in the WB bands shown.
- We are not quite sure what the Reviewer is referring to here. We do not state that we are observing a significant reduction in the levels of LC3I in Septin5+/- or Septin5-/- mice, but that there is a trend towards this. This trend is fairly evident in the case of Septin5-/- mice, and quantitation of the Western blots reveals a ~25% reduction in LC3I levels for Septin5-/- mice, when compared to Septin5+/+ mice. However, this reduction did not reach statistical difference and should only be interpreted as an implication of a potential decrease in LC3I levels. This has now been indicated in the manuscript text (line 443).
References
- Kurkinen, K.M.A.; Marttinen, M.; Turner, L.; Natunen, T.; Mäkinen, P.; Haapalinna, F.; Sarajärvi, T.; Gabbouj, S.; Kurki, M.; Paananen, J.; et al. SEPT8 modulates β-amyloidogenic processing of APP by affecting the sorting and accumulation of BACE1. J. Cell Sci. 2016, 129, 2224–2238, doi:10.1242/jcs.185215.
- Natunen, T.; Takalo, M.; Kemppainen, S.; Leskelä, S.; Marttinen, M.; Kurkinen, K.M.A.; Pursiheimo, J.-P.; Sarajärvi, T.; Viswanathan, J.; Gabbouj, S.; et al. Relationship between ubiquilin-1 and BACE1 in human Alzheimer’s disease and APdE9 transgenic mouse brain and cell-based models. Neurobiol. Dis. 2016, 85, doi:10.1016/j.nbd.2015.11.005.
- Viswanathan, J.; Haapasalo, A.; Böttcher, C.; Miettinen, R.; Kurkinen, K.M.A.; Lu, A.; Thomas, A.; Maynard, C.J.; Romano, D.; Hyman, B.T.; et al. Alzheimer’s Disease-Associated Ubiquilin-1 Regulates Presenilin-1 Accumulation and Aggresome Formation. Traffic 2011, 12, 330–348, doi:10.1111/j.1600-0854.2010.01149.x.
- Storey, E. Amyloid precursor protein of Alzheimer’s disease: Evidence for a stable, full-length, trans-membrane pool in primary neuronal cultures. Eur. J. Neurosci. 1999, doi:10.1046/j.1460-9568.1999.00599.x.
Reviewer 2 Report
Generally, this is a well written manuscript, using concise language.
We have the following comments:
Material & Methods
Section 2.2: It is no clear whether the vector encodes a GFP-LC3 fusion protein or if both proteins are produced and released independently. This is important for the interpretation of data presented in Figure 5, so please provide more details on the vector and its products.
Section 2.10: The description of immunofluorescence is confusing and probably contains a few errors. While only incubations with two different rabbit primary antibodies are mentioned in lines 214-216 (SEPTIN5 and APP C-terminus; consistent with the images in Figure 5) visualization with both anti-rabbit and anti-mouse secondary antibodies are described in lines 217-218. In addition, line 217 says the anti-mouse antibody was conjugated with Alexa Fluor 488, which would interefere with the signal of GFP-LC3. Please correct or provide more information to understand what was done.
Results
Figure 1 and 2: Please mention ELISA somewhere in legend or description as the source of the Abeta40/42 data. The only place this information is currently available is in the Material and Methods.
Figure 2D: Curiously, only data on Abeta42 are presented here, while Abeta40 is missing. However, data on both types are presented in Fig. 1B and Fig. 2B. These data should be added; at least there should be an explanation or some kind of comment on this in the figure legend or description.
Figure 3D: Steadily decreasing levels of APPtot in both groups, but no change at all for APP CTFs in siControl: A comment on this difference should be provided, because it is of interest for interpretation of the result.
Figure 4: Text in line 439-440: Wrong reference to panel 4a (should be 4C); statistical result described ("... levels of LC3I were significantly decreased already at time point 0 min ...") but missing from figure; please present data or rephrase the desciption.
Figure 5: Panel A; we believe the white arrow is not in the correct position intended by the authors; please check. Panel C; why is there so much GFP signal in nuclei? Was a GFP-LC3 fusion protein used? If not, how does this affect the validity of the data? Please comment. It is obvious from the single channel images in panel C that GFP-LC3 signal is much lower in the bottom images [group siSSEPTIN5/GFP-LC3 (BfA1 2h)]. If this is representative the low signal would probably explain most of the lower % colocalization of APP CTFs and GFP-LC3 in BfA1-treated cells (graph on the bottom right in Figure 5); this should be mentioned and discussed.
Minor findings:
The term "ventral cortex" is used repeatedly throughout the manuscript (for example in lines 236, 348 and 545); is this orbital frontal cortex, piriform cortex, cortical amygdala, lateral entorhinal cortex, or all of the above? Please specify how the tissue was collected and which major regions are roughly included using established nomeclature such as those used in the Paxinos or Allen mouse brain atlases.
Please use consistent terminology, writing style, and formatting throughout the manuscript; currently both SEPTIN5 and SEPT5 (lines 78-79) are used; Septin5 knock-out (line 91) and Septin5 knockout (line 126); see also genotypes in lines 319-331 and in the legend of Fig. 2.
Add host species (rabbit) to SEPTIN5 antibody in line 164.
Change Nonident to Nonidet in line 194.
The term "time point" is used frquently in the manuscript but does not really exist in English; please replace by "at time 0 min" or "at 0 min"
Type setting error: Figure titles are in italics in legends of figures 4 and 5.
Author Response
Reviewer 2
Overall, the characterization of cellular and molecular mechanisms underlying SEPTINs control of APP is certainly of interest for a deeper understanding of Alzheimer’s Disease etiopathology. However, the effect of SEPTIN5 seams to also extend to BACE1, and potentially to other not investigated targets, arguing its relevance as putative therapeutic target.
Generally, this is a well written manuscript, using concise language.
- We thank the Reviewer for the comments. First, we would like to emphasize that our data suggest BACE1-independent effects on APP, mainly being conveyed via altered autophagosomal degradation of APP C-terminal fragments. We only observed alterations in the levels of BACE1 protein in Septin5 knockout mice, but not in SH-SY5Y-APP751 cells or cultured primary neurons, making this an inconsistent result. Whether BACE1 might show differential regulation in vitro and in vivo would require further clarification in a separate study. Below we provide point-by-point answers to the Reviewer’s comments.
Section 2.2: It is no clear whether the vector encodes a GFP-LC3 fusion protein or if both proteins are produced and released independently. This is important for the interpretation of data presented in Figure 5, so please provide more details on the vector and its products.
- This is a relevant point. We have now clarified in the Materials and Methods that the GFP-LC3 plasmid produces a fusion protein, suitable for colocalization analysis. The same construct has been used in fluorescence microscopy of autophagosomal vesicles in our previously published study [1].
Section 2.10: The description of immunofluorescence is confusing and probably contains a few errors. While only incubations with two different rabbit primary antibodies are mentioned in lines 214-216 (SEPTIN5 and APP C-terminus; consistent with the images in Figure 5) visualization with both anti-rabbit and anti-mouse secondary antibodies are described in lines 217-218. In addition, line 217 says the anti-mouse antibody was conjugated with Alexa Fluor 488, which would interefere with the signal of GFP-LC3. Please correct or provide more information to understand what was done.
- The Reviewer is correct, this is a mistake and we thank the reviewer for pointing this out. Only Alexa Fluor 594 goat anti-rabbit antibody was used for staining. Remarks to Alexa Fluor 488 anti-mouse have been now removed.
Figure 1 and 2: Please mention ELISA somewhere in legend or description as the source of the Abeta40/42 data. The only place this information is currently available is in the Material and Methods.
- This is a good point, which was left unclear in the original version of the manuscript and we apologize for this. Ab40 and Ab42 levels were measured by ELISA, and has now been clarified in the relevant figure captions. An example from Figure 1B caption:
“B) Levels of sAPPβ (Western blot) and Aβ (ELISA) measured from cell culture medium indicated a decrease upon SEPTIN5 downregulation. The levels of sAPPtot and APPα remained unchanged.”
Figure 2D: Curiously, only data on Abeta42 are presented here, while Abeta40 is missing. However, data on both types are presented in Fig. 1B and Fig. 2B. These data should be added; at least there should be an explanation or some kind of comment on this in the figure legend or description.
- Thank you for pointing this out. In fact, this had already been mentioned in the main text as follows: Lines 334-335 “The levels of sAPPβ and Aβ40 were below detection limits in these samples.”
Figure 3D: Steadily decreasing levels of APPtot in both groups, but no change at all for APP CTFs in siControl: A comment on this difference should be provided, because it is of interest for interpretation of the result.
- We agree with the Reviewer that this should be discussed. We believe this is a by-product of cycloheximide. It has been previously shown that cycloheximide can also moderately inhibit autophagy. Given that APP CTFs are primarily degraded via autophagy, this could result in stabilization of APP CTFs in the case of siControl siRNA-transfected samples. However, the effect of SEPTIN5 downregulation seems to be robust enough to overcome this moderate inhibition. We have added this discussion to the text as follows:
Lines 392-396
“The levels of APP CTFs remained unaltered and were close to 100% in siControl siRNA-transfected cells throughout the 80-min time course. This observation is potentially related to the fact that APP CTFs are primarily degraded via the autophagosomal-lysosomal degradation pathway, and that cycloheximide, in addition to blocking de novo protein synthesis, also moderately inhibits the autophagosomal-lysosomal pathway [23,24].
Figure 4: Text in line 439-440: Wrong reference to panel 4a (should be 4C); statistical result described ("... levels of LC3I were significantly decreased already at time point 0 min ...") but missing from figure; please present data or rephrase the desciption.
- We agree with the Reviewer that this requires reformatting. We have now removed the reference to Figure 4A as time-point 0, and we merely mention that in a specific timepoint snapshot (as done in Figure 4A), we observe a significant downregulation in LC3I protein levels in SEPTIN5 siRNA-transfected samples as compared to siControl-transfected samples. Additionally, when performing a cycloheximide time-course assay, we observed an accelerated degradation rate for LC3I in SEPTIN5 siRNA-transfected samples as compared to siControl-transfected samples. This issue has now been added to the text as indicated below:
Lines 446-451
“To delineate whether the observed decrease in the levels of LC3 upon downregulation of SEPTIN5 was related to changes in expression or degradation, the de novo protein synthesis was blocked by cycloheximide during an 80-min time course in SH-SY5Y-APP751 cells (Figure 4C). LC3I was degraded significantly faster during the 80-min time course in SEPTIN5 siRNA-transfected cells as compared to control siRNA-transfected cells (Figure 4C). This suggests that the degradation rather than the expression of LC3 is enhanced due to the downregulation of SEPTIN5 in SH-SY5Y-APP751 cells.”
Figure 5: Panel A; we believe the white arrow is not in the correct position intended by the authors; please check. Panel C; why is there so much GFP signal in nuclei? Was a GFP-LC3 fusion protein used? If not, how does this affect the validity of the data? Please comment. It is obvious from the single channel images in panel C that GFP-LC3 signal is much lower in the bottom images [group siSSEPTIN5/GFP-LC3 (BfA1 2h)]. If this is representative the low signal would probably explain most of the lower % colocalization of APP CTFs and GFP-LC3 in BfA1-treated cells (graph on the bottom right in Figure 5); this should be mentioned and discussed.
- The white arrow in Figure 5, Panel A is in the correct position. The arrow shows the start and end positions of the histogram shown below, which portrays red and green fluorescence signals across the observed vesicle-like structure, which most likely is an autophagosomal vesicle given the puncta-like GFP-LC3 signal. To clarify this, we have now modified the Figure 5 caption as follows: “Quantification of the overlapping fluorescence intensity for a GFP-LC3 positive vesicle (across the white arrow) indicated marked co-localization for SEPTIN5 and GFP-LC3.”
- As answered above, a GFP-LC3 fusion protein was used, and this issue has been now clarified in the Material and Methods part.
- Given that we are looking at proportions between groups in Figure 5, panel C, the overall level of a protein should not affect this. In this case, would we expect to see a smaller proportion of GFP-LC3 colocalizing with APP-CTFs when there is less GFP-LC3? We argue that we would not, as the equation to calculate this would be as follows: . Hence if you modulate LC3 signal you are affecting both the numerator and denominator, whereas modulating APP-CTF levels would only affect the numerator. We have discussed this point in the manuscript as follows on lines 484-487: “However, in line with the Western blot data in Figure 4A, the downregulation of SEPTIN5 resulted in a decreased proportion of GFP-LC3 co-localized with APP CTFs, which is most likely due to the reduced levels of APP CTFs in these cells (Figure 5C).”
The term "ventral cortex" is used repeatedly throughout the manuscript (for example in lines 236, 348 and 545); is this orbital frontal cortex, piriform cortex, cortical amygdala, lateral entorhinal cortex, or all of the above? Please specify how the tissue was collected and which major regions are roughly included using established nomeclature such as those used in the Paxinos or Allen mouse brain atlases.
- Additional clarification to the dissection procedure has been now provided in the Material and Methods art as stated below. The term ventral cortex has now been accordingly substituted with the term temporal cortex.
“For dissection of tissue samples, three coronal cuts were made using a blade approximately at bregma 2.00mm, 0.00mm and -5.00mm. The frontal cortex was collected after the first cut. The motor cortex and the striatum were collected after the second cut. The hippocampus and the posterior cortex were collected after the third cut by removing the midbrain. The posterior cortex was cut in the midline to generate the dorsal and temporal cortex. The ventral posterior cortex was used for Western blot and ELISA analysis.”
Please use consistent terminology, writing style, and formatting throughout the manuscript; currently both SEPTIN5 and SEPT5 (lines 78-79) are used; Septin5 knock-out (line 91) and Septin5 knockout (line 126); see also genotypes in lines 319-331 and in the legend of Fig. 2.
- This has been corrected as suggested.
Add host species (rabbit) to SEPTIN5 antibody in line 164.
- This has been corrected as suggested.
Change Nonident to Nonidet in line 194.
- This has been corrected as suggested.
The term "time point" is used frquently in the manuscript but does not really exist in English; please replace by "at time 0 min" or "at 0 min"
- This has been corrected as suggested.
Type setting error: Figure titles are in italics in legends of figures 4 and 5.
- This has been now corrected.
References
- Leskelä, S.; Huber, N.; Rostalski, H.; Natunen, T.; Remes, A.M.; Takalo, M.; Hiltunen, M.; Haapasalo, A. C9orf72 Proteins Regulate Autophagy and Undergo Autophagosomal or Proteasomal Degradation in a Cell Type-Dependent Manner. Cells 2019, doi:10.3390/cells8101233.
Reviewer 2
Overall, the characterization of cellular and molecular mechanisms underlying SEPTINs control of APP is certainly of interest for a deeper understanding of Alzheimer’s Disease etiopathology. However, the effect of SEPTIN5 seams to also extend to BACE1, and potentially to other not investigated targets, arguing its relevance as putative therapeutic target.
Generally, this is a well written manuscript, using concise language.
- We thank the Reviewer for the comments. First, we would like to emphasize that our data suggest BACE1-independent effects on APP, mainly being conveyed via altered autophagosomal degradation of APP C-terminal fragments. We only observed alterations in the levels of BACE1 protein in Septin5 knockout mice, but not in SH-SY5Y-APP751 cells or cultured primary neurons, making this an inconsistent result. Whether BACE1 might show differential regulation in vitro and in vivo would require further clarification in a separate study. Below we provide point-by-point answers to the Reviewer’s comments.
Section 2.2: It is no clear whether the vector encodes a GFP-LC3 fusion protein or if both proteins are produced and released independently. This is important for the interpretation of data presented in Figure 5, so please provide more details on the vector and its products.
- This is a relevant point. We have now clarified in the Materials and Methods that the GFP-LC3 plasmid produces a fusion protein, suitable for colocalization analysis. The same construct has been used in fluorescence microscopy of autophagosomal vesicles in our previously published study [1].
Section 2.10: The description of immunofluorescence is confusing and probably contains a few errors. While only incubations with two different rabbit primary antibodies are mentioned in lines 214-216 (SEPTIN5 and APP C-terminus; consistent with the images in Figure 5) visualization with both anti-rabbit and anti-mouse secondary antibodies are described in lines 217-218. In addition, line 217 says the anti-mouse antibody was conjugated with Alexa Fluor 488, which would interefere with the signal of GFP-LC3. Please correct or provide more information to understand what was done.
- The Reviewer is correct, this is a mistake and we thank the reviewer for pointing this out. Only Alexa Fluor 594 goat anti-rabbit antibody was used for staining. Remarks to Alexa Fluor 488 anti-mouse have been now removed.
Figure 1 and 2: Please mention ELISA somewhere in legend or description as the source of the Abeta40/42 data. The only place this information is currently available is in the Material and Methods.
- This is a good point, which was left unclear in the original version of the manuscript and we apologize for this. Ab40 and Ab42 levels were measured by ELISA, and has now been clarified in the relevant figure captions. An example from Figure 1B caption:
“B) Levels of sAPPβ (Western blot) and Aβ (ELISA) measured from cell culture medium indicated a decrease upon SEPTIN5 downregulation. The levels of sAPPtot and APPα remained unchanged.”
Figure 2D: Curiously, only data on Abeta42 are presented here, while Abeta40 is missing. However, data on both types are presented in Fig. 1B and Fig. 2B. These data should be added; at least there should be an explanation or some kind of comment on this in the figure legend or description.
- Thank you for pointing this out. In fact, this had already been mentioned in the main text as follows: Lines 334-335 “The levels of sAPPβ and Aβ40 were below detection limits in these samples.”
Figure 3D: Steadily decreasing levels of APPtot in both groups, but no change at all for APP CTFs in siControl: A comment on this difference should be provided, because it is of interest for interpretation of the result.
- We agree with the Reviewer that this should be discussed. We believe this is a by-product of cycloheximide. It has been previously shown that cycloheximide can also moderately inhibit autophagy. Given that APP CTFs are primarily degraded via autophagy, this could result in stabilization of APP CTFs in the case of siControl siRNA-transfected samples. However, the effect of SEPTIN5 downregulation seems to be robust enough to overcome this moderate inhibition. We have added this discussion to the text as follows:
Lines 392-396
“The levels of APP CTFs remained unaltered and were close to 100% in siControl siRNA-transfected cells throughout the 80-min time course. This observation is potentially related to the fact that APP CTFs are primarily degraded via the autophagosomal-lysosomal degradation pathway, and that cycloheximide, in addition to blocking de novo protein synthesis, also moderately inhibits the autophagosomal-lysosomal pathway [23,24].
Figure 4: Text in line 439-440: Wrong reference to panel 4a (should be 4C); statistical result described ("... levels of LC3I were significantly decreased already at time point 0 min ...") but missing from figure; please present data or rephrase the desciption.
- We agree with the Reviewer that this requires reformatting. We have now removed the reference to Figure 4A as time-point 0, and we merely mention that in a specific timepoint snapshot (as done in Figure 4A), we observe a significant downregulation in LC3I protein levels in SEPTIN5 siRNA-transfected samples as compared to siControl-transfected samples. Additionally, when performing a cycloheximide time-course assay, we observed an accelerated degradation rate for LC3I in SEPTIN5 siRNA-transfected samples as compared to siControl-transfected samples. This issue has now been added to the text as indicated below:
Lines 446-451
“To delineate whether the observed decrease in the levels of LC3 upon downregulation of SEPTIN5 was related to changes in expression or degradation, the de novo protein synthesis was blocked by cycloheximide during an 80-min time course in SH-SY5Y-APP751 cells (Figure 4C). LC3I was degraded significantly faster during the 80-min time course in SEPTIN5 siRNA-transfected cells as compared to control siRNA-transfected cells (Figure 4C). This suggests that the degradation rather than the expression of LC3 is enhanced due to the downregulation of SEPTIN5 in SH-SY5Y-APP751 cells.”
Figure 5: Panel A; we believe the white arrow is not in the correct position intended by the authors; please check. Panel C; why is there so much GFP signal in nuclei? Was a GFP-LC3 fusion protein used? If not, how does this affect the validity of the data? Please comment. It is obvious from the single channel images in panel C that GFP-LC3 signal is much lower in the bottom images [group siSSEPTIN5/GFP-LC3 (BfA1 2h)]. If this is representative the low signal would probably explain most of the lower % colocalization of APP CTFs and GFP-LC3 in BfA1-treated cells (graph on the bottom right in Figure 5); this should be mentioned and discussed.
- The white arrow in Figure 5, Panel A is in the correct position. The arrow shows the start and end positions of the histogram shown below, which portrays red and green fluorescence signals across the observed vesicle-like structure, which most likely is an autophagosomal vesicle given the puncta-like GFP-LC3 signal. To clarify this, we have now modified the Figure 5 caption as follows: “Quantification of the overlapping fluorescence intensity for a GFP-LC3 positive vesicle (across the white arrow) indicated marked co-localization for SEPTIN5 and GFP-LC3.”
- As answered above, a GFP-LC3 fusion protein was used, and this issue has been now clarified in the Material and Methods part.
- Given that we are looking at proportions between groups in Figure 5, panel C, the overall level of a protein should not affect this. In this case, would we expect to see a smaller proportion of GFP-LC3 colocalizing with APP-CTFs when there is less GFP-LC3? We argue that we would not, as the equation to calculate this would be as follows: . Hence if you modulate LC3 signal you are affecting both the numerator and denominator, whereas modulating APP-CTF levels would only affect the numerator. We have discussed this point in the manuscript as follows on lines 484-487: “However, in line with the Western blot data in Figure 4A, the downregulation of SEPTIN5 resulted in a decreased proportion of GFP-LC3 co-localized with APP CTFs, which is most likely due to the reduced levels of APP CTFs in these cells (Figure 5C).”
The term "ventral cortex" is used repeatedly throughout the manuscript (for example in lines 236, 348 and 545); is this orbital frontal cortex, piriform cortex, cortical amygdala, lateral entorhinal cortex, or all of the above? Please specify how the tissue was collected and which major regions are roughly included using established nomeclature such as those used in the Paxinos or Allen mouse brain atlases.
- Additional clarification to the dissection procedure has been now provided in the Material and Methods art as stated below. The term ventral cortex has now been accordingly substituted with the term temporal cortex.
“For dissection of tissue samples, three coronal cuts were made using a blade approximately at bregma 2.00mm, 0.00mm and -5.00mm. The frontal cortex was collected after the first cut. The motor cortex and the striatum were collected after the second cut. The hippocampus and the posterior cortex were collected after the third cut by removing the midbrain. The posterior cortex was cut in the midline to generate the dorsal and temporal cortex. The ventral posterior cortex was used for Western blot and ELISA analysis.”
Please use consistent terminology, writing style, and formatting throughout the manuscript; currently both SEPTIN5 and SEPT5 (lines 78-79) are used; Septin5 knock-out (line 91) and Septin5 knockout (line 126); see also genotypes in lines 319-331 and in the legend of Fig. 2.
- This has been corrected as suggested.
Add host species (rabbit) to SEPTIN5 antibody in line 164.
- This has been corrected as suggested.
Change Nonident to Nonidet in line 194.
- This has been corrected as suggested.
The term "time point" is used frquently in the manuscript but does not really exist in English; please replace by "at time 0 min" or "at 0 min"
- This has been corrected as suggested.
Type setting error: Figure titles are in italics in legends of figures 4 and 5.
- This has been now corrected.
References
- Leskelä, S.; Huber, N.; Rostalski, H.; Natunen, T.; Remes, A.M.; Takalo, M.; Hiltunen, M.; Haapasalo, A. C9orf72 Proteins Regulate Autophagy and Undergo Autophagosomal or Proteasomal Degradation in a Cell Type-Dependent Manner. Cells 2019, doi:10.3390/cells8101233.